# INRscrecon: Enhancing 3D Spatial Transcriptomics Reconstruction through Implicit Neural Representations

## Abstract

Single-cell spatial transcriptomics (scST) technologies have revolutionized our understanding of the complex three-dimensional cellular landscapes of tissues. However, the accuracy of spatial expression profiles is often compromised by missing or distorted experimental data. To address this challenge, we introduce INRscrecon, a novel framework that leverages Implicit Neural Representations (INRs) known for their continuous signal encoding capabilities. INRscrecon accurately predicts and corrects spatial expressions, enhancing the clarity of 3D tissue reconstructions. Our study demonstrates the efficacy of INRscrecon across various datasets and dimensions, highlighting its potential to restore spatial expression with high precision. The findings suggest broader applications for INR-based methodologies in spatial transcriptomics, paving the way for more accurate and detailed analysis of cellular interactions within tissues. Future research may expand on the incorporation of INR techniques in spatial transcriptomics to further enhance analytical capabilities.

## 1 Introduction

Tissues consist of different cells arranged into specific functional areas, each with unique cell types and diverse gene expression profiles(Wang et al., 2022a; Rao et al., 2021; Moses & Pachter, 2022). Single-cell spatial transcriptomics (scST), named 'Method of the Year' by Nature Methods in 2020(Marx, 2021), enables gene expression measurement while preserving the spatial location information at sites like spots, beads, or peaks. Figure 1 displays the various technologies available in scST. This technology enables the measurement of gene expression within the native tissue context, allowing researchers to characterize patterns of spatial gene expression, investigate cell-cell communication, and elucidate the spatiotemporal dynamics of cellular development(Tian et al., 2023; Burgess, 2019; Bressan et al., 2023; Moffitt et al., 2022b).

However, accurate profile scST in 3D tissue faces numerous technical challenges. These include cross-contamination of transcripts or data loss, which typically result in a significant number of outliers in the data(Zeira et al., 2022a; Walker & Nie, 2023; Atta & Fan, 2021). Additionally, each spatial location may contain tens of thousands of dimensions of gene expression data. Coupled with the extreme sparsity of measurement points, low spatial resolution, and overall poor data quality, these factors collectively obscure the continuous expression patterns of many genes in space. This severely affects the accurate retrieval of expression values, thereby having a major impact on the construction and analysis of 3D tissue atlases(Velten et al., 2022; Townes & Engelhardt, 2023). Therefore, designing algorithms to restore the high-dimensional gene expression information in space and recover the spatial continuity of gene expression has become a crucial challenge in solving problems in spatial transcriptomics.

Leveraging the potential of Multi-Layer Perceptrons (MLPs) to fit continuous functions known as Implicit Neural Representations (INRs), combined with the theory of continuous gene expression, we developed the INRscrecon specifically for reconstructing spatial transcriptomics data in 3D(Xie et al., 2022; Yang et al., 2021; Darmon et al., 2022; Liu et al., 2020). This method encodes gene expression in space through MLP-parameterized continuous functions, thereby decoupling the memory cost from the actual spatial resolution and allowing the reconstruction of expression at any desired resolution using a fixed number of parameters. The main features of this algorithm include: (i)

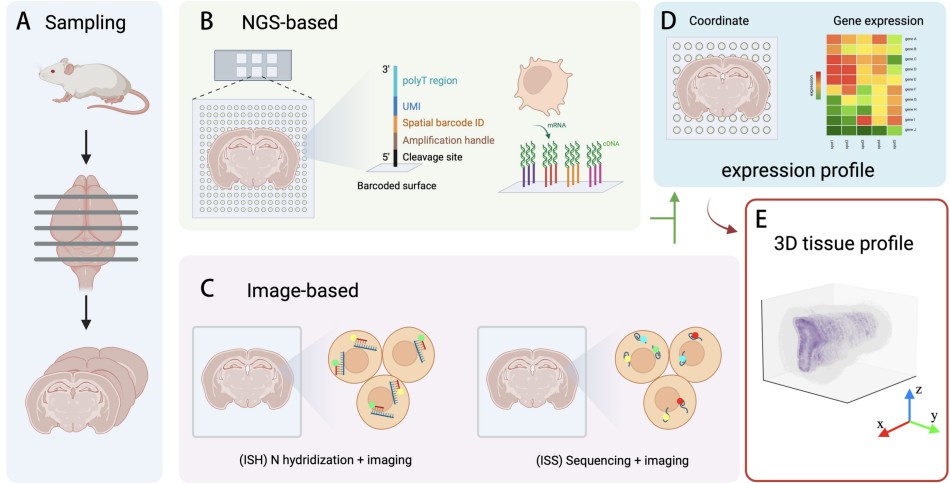

**Figure 1: Overall workflow for 3D reconstruction of serial sections of the spatial transcriptome** A, The spatial transcriptome was used to sample the embedded tissues, and the tissue samples were cut into thin sections using a microtome. B, Based on the strategy of Next-Generation Sequencing (NGS), short sequences with spatial information (spatial barcode/ID) are attached to reads from different sources. C, Based on the Image strategy, this strategy is mainly completed by in-situ amplification and in-situ sequencing and in situ hybridization. D, The ultimate goal of these strategies is to obtain coordinate information and gene matrix. E, By 3D reconstruction of spatial transcriptome data from serial sections, 3D tissue expression profile can be obtained to aid biological discovery. More information can be found in appendix.

precise spatial analysis through coordinate-based fitting of INRs, enhancing the identification of cell types and their spatial distributions; (ii) 3D spatial reconstruction by restoring the continuity of gene expression, thereby more effectively restoring detailed expression patterns. Through multiple evaluations and benchmarks, we demonstrate that INRscrecon is capable of predicting missing tissue sections and successfully reconstructing continuous tissue atlases. Compared to previous methods, INRscrecon exhibits significant potential in spatial transcriptomics applications due to its robust handling of data sparsity and continuity in tissue representation.

## 2 RELATED WORK

### 2.1 3D MOLECULAR PROFILE METHODS OF SPATIAL TRANSCRIPTOMICS

Spatial profiling in 3D presents a significant challenge for scST. Firstly, accurately aligning multiple tissue slices is a significant challenge, often involving the use of thin tissue sections to generate spatial molecular data. The insights derived from 2D spatial profiling are incomplete, offering merely a snapshot that captures only a fragment of the complex spatial context of tissue structures. Secondly, building sufficient resolution is constrained by the limited spatial resolution available, which hinders our ability to deeply characterize intricate tissue architectures(Xiao et al., 2024; Moffitt et al., 2022a; Coleman et al., 2024; Wang et al., 2024).

To achieve precise alignment of high-resolution single-cell data across sequential slides, several advanced methods have been employed: GPSA(Jones et al., 2023), a Bayesian model, aligns spatial genomic and histology samples, compensating for distortions or system differences with unique warping functions for each slice. STalign(Clifton et al., 2023) builds on advancements in Large Deformation Diffeomorphic Metric Mapping (LDDMM) using image varifolds to align ST datasets. PASTE(Zeira et al., 2022b) uses probabilistic pairwise alignments and fused Gromov-Wasserstein optimal transport to enhance accuracy based on transcriptional and spatial similarities. ST-GEARS(Xia et al., 2024) advances this approach by incorporating bi-sectional applications to improve predictive accuracy. Graph neural networks like STAGATE(Dong & Zhang, 2022), STAIR(Yu & Xie, 2024), and Stitch3D(Wang et al., 2023) are increasingly used to enhance alignment and predictive accuracy in spatial transcriptomics. Additionally, integrating various multi-omics data or multiple scST technology can enhance resolution and alignment efficiency for constructing 3D tissue profiles. soScope(Li et al., 2024a) leverages a multimodal deep learning frame-

work to integrate spot omics profiles, spatial relationships, and high-resolution morphological images, jointly inferring omics profiles at subspot resolution. This model effectively reduces variation across datasets by selecting omics-specific distributions. SLAT(Xia et al., 2023) can align heterogeneous spatial data across different technologies and modalities. FuseMap(He et al., 2024) combines various spatial transcriptomics technologies using a self-supervised learning approach to generate universal representations of genes, cells, and tissues, thereby eliminating the need for existing cell annotations. SPACEL(Xu et al., 2023) is a deep-learning toolkit with three modules—Spoint, Splane, and Scube—that collectively enable the stacking of multiple consecutive slices to construct a comprehensive 3D architecture of tissue.

In reconstructing 3D tissue structures, current methods typically use discrete representations and often fail to predict missing sections between tissue samples. Most models overlook these gaps, except for the STAGE(Li et al., 2024b) model, which tries to predict these intermediate areas. However, INRs offer a promising solution due to their inherent continuity, enabling them to fill in missing parts in spatial transcriptomics data effectively. This capability provides a novel and practical approach for improving tissue reconstruction in research.

## 2.2 IMPLICIT NEURAL REPRESENTATION IN BIOLOGY

Implicit Neural Representations (INRs), powered by deep learning algorithms, encode signals within their weights, facilitating the transformation of discrete representations into a continuous spatial domain. This capability greatly enhances their applicability across various biomedical tasks, thereby broadening researchers' understanding of the significant functionalities and operations of these models(Chen et al., 2023). Initially, Molaei et al. (2023a) offers a comprehensive summary of INR applications in the medical field. For instance, the NeRP(Shen et al., 2022) framework proposes integrating implicit neural networks to reconstruct sparsely sampled medical images across three stages without requiring any training data. Another approach, DCTR(Fei et al., 2023), uses computed tomography (4D-CT) to reconstruct dynamic, time-varying scenes by employing INRs to estimate a template reconstruction of the 3D volume's linear attenuation coefficients (LACs), serving as a prior model that captures the spatial distribution of LACs. The mirnf(Molaei et al., 2023b) model can simulate both displacement vector fields and velocity vector fields, offering two distinct methods for image registration: displacement vector fields for deformable registration and velocity vector fields for diffeomorphic registration, both utilizing INRs to model transformations between the target and moving images. Additionally, TiAVox(Zhou et al., 2023) is a method for reconstructing 4D digital subtraction angiography (4D DSA) from sparse views based on time-aware attenuated voxels, reducing radiation dosage while achieving high-quality 4D imaging, capable of generating both 2D and 3D DSA images. These methods leverage the continuous implicit fields of INRs for high-definition alignment and reconstruction. Furthermore, target-data-specific approaches like INR have shown promise in effectively compressing diverse visual data. TINC(Yang, 2023), for instance, uses MLPs to fit segmented local regions arranged in a tree structure to enable parameter sharing based on spatial distance. Experiments on the HiP-CT dataset demonstrate TINC's superiority over conventional techniques, although it has slower compression speeds, its decompression speed is high.

However, these approaches often overlook micro-scale biological elements such as protein structure reconstruction. Recognizing the potential of INRs for both macro and micro-biological applications, Cai et al. (2024) highlighted advancements with CryoDRGN(Zhong et al., 2021), which marks a significant breakthrough in cryo-EM data processing and 3D structural reconstruction. CryoDRGN employs a INR-based deep learning architecture to facilitate ab initio unsupervised reconstruction of continuous distribution 3D volumes directly from unlabeled 2D images. Additionally, CryoDRGN-ET(Rangan et al., 2024) was developed to visualize the structures of dynamic macromolecules in native cellular environments. While cryo-ET can achieve molecular resolution, its image processing algorithms are further enhanced by INR capabilities. Given the substantial demand for alignment and reconstruction in spatial transcriptomics, INRs possess significant potential to revolutionize this field.

## 3 METHODS

We propose an method named INRscrecon (Figure 2), which consists of two main components: the Alignment component and the Imputation component. The Alignment component can be regarded as a data preprocessing step, where two sets of single-cell coordinates are aligned pairwise. The Imputation component focuses on learning the mapping between cell coordinates and gene expression profiles, subsequently imputing a missing slide based on the learned spatial patterns.

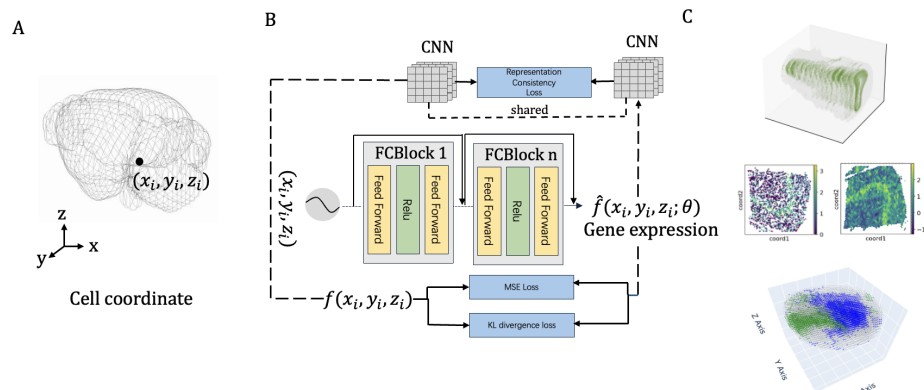

**Figure 2: Overview of the INRscrecon Algorithm Framework** In the INRscrecon model, part A takes tissue coordinates as input; part B employs an alignment component and an imputation component to learn and refine the data; ultimately, part C outputs the gene expression data.

## 3.1 ALIGNMENT

To align two spatial coordinates (SC) from single-cell data, we seek to find a transformation function $\Phi$ that aligns the coordinates x in SC1 with the coordinates y in SC2. In the rigid alignment problem, we assume that SC2 is derived from SC1 through an unknown rigid motion. We define the rigid transformation as $[R; t]$, where $R \in \mathbb{R}^2$ is the rotation matrix and $t \in \mathbb{R}^2$ is the translation vector. Thus, the transformation can be expressed as $\Phi(x) = Rx + t$. Our objective is to find the optimal transformation $\Phi^*$, formulated as:

$$\Phi^* = \arg\min_{\hat{\Phi}} \left( L_{\text{data}}(SC1 \circ \hat{\Phi}, SC2) \right) \tag{1}$$

To compute the transformation, we use the gene expression data $\{SC1_{\text{gene}}, SC2_{\text{gene}}\}$ to calculate the correlation coefficient, which allows us to match the coordinates $SC1_{\text{coord}} \mapsto SC2_{\text{coord}}$. We then construct the cross-covariance matrix H from the centroids of $SC1_{\text{coord}}$ and $SC2_{\text{coord}}$, and perform singular value decomposition (SVD) on H, where $H = USV^T$. The optimal rotation matrix R is given by $R = VU^T$, and the translation vector t is computed as $t = -R\bar{x} + \bar{y}$.

## 3.2 INRSCRECON MODEL

The INRscrecon model takes as input cell coordinates $C \in \mathbb{R}^3$ and predicts the corresponding gene expression profiles $G \in \mathbb{R}^M$, where M = 5000. Thus, the model aims to learn a mapping $\hat{SC} : \mathbb{R}^3 \rightarrow \mathbb{R}^M$, which translates spatial coordinates into gene expression profiles. The core of the model is a multilayer perceptron (MLP) comprising n layers. Before being fed into the MLP, the input cell coordinates undergo positional encoding(Tancik et al., 2020) to enhance the model's capacity to capture spatial dependencies present in the data.

$$\text{pe}_{2i}(c_j) = \sin\left(c_j \cdot D \cdot \frac{\pi(2/D)}{2i/D}\right), \quad i = 1, \ldots, D/2; \, c_j \in c \tag{2}$$

$$\text{pe}_{2i+1}(c_j) = \cos\left(c_j \cdot D \cdot \frac{\pi(2/D)}{2i/D}\right), \quad i = 1, \ldots, D/2; \, c_j \in c \tag{3}$$

To ensure accurate learning of the mapping between cell coordinates and gene expression, three distinct loss functions are utilized during the training process. These loss functions collectively guide the model to effectively learn and generalize the spatial-to-gene expression mapping. To ensure that the model effectively learns the mapping, we employ three loss functions during the training process:1. Reconstruction Loss: This loss function computes the root mean square error (RMSE) between the predicted gene expression and the ground truth gene expression, ensuring that the model accurately reconstructs the gene expression profiles. 2. Representation Consistency Loss: A convolutional neural network (CNN) is utilized to extract latent representations from the gene expression profiles, the detailed CNN architecture can be found in appendix. This loss function supervises

the alignment between the predicted latent features and the ground truth latent features, helping the model to capture intricate patterns within the gene expression data. 3.KL Divergence Loss: The Kullback-Leibler (KL) divergence loss is applied to minimize the distributional divergence between the predicted gene expression and the ground truth gene expression, encouraging the model to closely approximate the underlying data distribution.These elements are described in further details.

### 3.2.1 RECONSTRUCTION LOSS

Since our model is designed to learn the mapping function $\hat{SC} : C \rightarrow \hat{G}$, where $C$ represents the cell coordinates and $\hat{G}$ denotes the predicted gene expression profiles, we employ the Root Mean Squared Error (RMSE) as the reconstruction loss. This loss function compares the real gene expression $G$ with the predicted gene expression $\hat{G}$ and encourages the model to minimize the discrepancies between the two at the raw gene expression level.

The reconstruction loss is formally defined as:

$$L_{\text{Rec}}(\hat{G}, G) = \ell_2(\hat{G}, G) = \sqrt{\frac{1}{n} \sum_{i=1}^{n} (\hat{G}_i - G_i)^2} \tag{4}$$

### 3.2.2 REPRESENTATION CONSISTENCY LOSS

To enhance the model's ability to learn complex mappings at a deeper feature level, we introduce a feature extraction module designed to capture the higher-order relationships in the gene expression data. This module consists of a convolutional neural network (CNN) followed by a two-layer multilayer perceptron (MLP), forming the feature extractor $H$. The role of the feature extractor is to map the high-dimensional gene expression profiles $G \in \mathbb{R}^M$ into a more compact, meaningful representation $z \in \mathbb{R}^d$, where $d$ is the dimensionality of the learned feature space. Given the true gene expression data $G$, the feature extractor outputs a latent representation $z = H(G)$. To ensure consistency between the real and predicted gene expression, the predicted gene expression $\hat{G}$ is also passed through the same feature extractor, resulting in $\hat{z} = H(\hat{G})$. Importantly, the feature extractor $H$ shares weights between the two passes, ensuring that both real and predicted data are mapped into the same feature space. This feature extraction process is applied at the batch level, and can be formalized as follows:

$$L_{ReqCons}(\hat{G}, G) = E_{\hat{g} \sim \hat{G}, g \sim G} \sum_{i=1}^{B} \ell_2(\hat{z}_i, z_i) \tag{5}$$

### 3.2.3 KULLBACK-LEIBLER DIVERGENCE LOSS

We additionally incorporate the Kullback-Leibler (KL) divergence as an auxiliary loss to further guide the model in capturing the relationship between the real and predicted gene expression distributions. By minimizing the KL divergence, the model is encouraged to align the predicted gene expression distribution with the real distribution, ensuring closer similarity between the two. This regularization promotes consistency between the distributions and helps the model to better generalize in its predictions.

$$L_{\text{KL}}(\hat{G} \parallel G) = \sum_i \hat{G}(i) \log\left(\frac{\hat{G}(i)}{G(i)}\right) \tag{6}$$

Our final objective is as follows.

$$L_{\text{Rec}}(\hat{G}, G) + \lambda_1 L_{ReqCons}(\hat{G}, G) + \lambda_2 L_{\text{KL}}(\hat{G} \parallel G) \tag{7}$$

## 4 RESULTS

To evaluate the performance of the INRscrecon model in spatial data 3D profiling, this study utilized datasets from Slide-seq, 10x Visium platform, and Stereo-seq, and compared them with existing spatial transcriptomics reconstruction algorithms such as STAGE and STitch3D. We systematically assessed these methods for their capabilities in spatial domain reconstruction, particularly highlighting

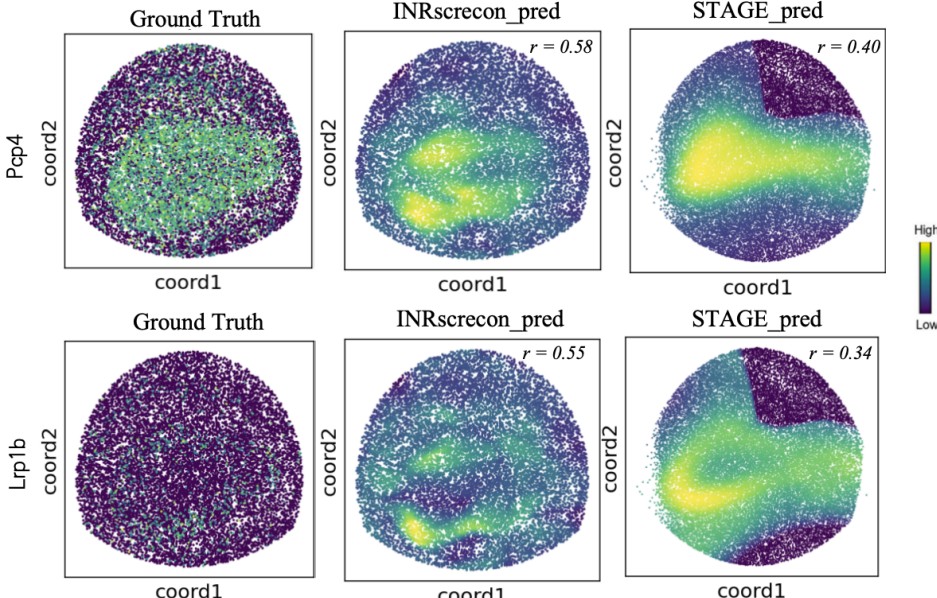

**Figure 3: Distribution Patterns of Genes** By randomly selecting genes, we examine their distribution on missing sections, comparing the actual versus predicted gene expressions using the STAGE and INRscrecon models, we use a single gene pattern as the ground truth, with yellow indicating high gene expression. We aim to predict and observe this pattern.

their ability to enhance spatial cell recognition and the detailed recovery of gene expression continuity. The INRscrecon demonstrated significant advantages in spatial domain identification and detail restoration. Through ablation studies, we further analyzed the specific contributions of each component within the model. These findings underscore INRscrecon's effectiveness in accurately restoring gene expression details, showcasing INRs substantial potential for biomedical research applications.

## 4.1 DATASETS AND BASELINES

To evaluate the effectiveness of INRscrecon in reconstructing spatial transcriptomic data, we conducted experiments on multiple publicly available datasets, each representing diverse biological contexts and technologies:

**Mouse Olfactory Bulb (OB) Dataset**. This dataset comprises eight slices of mouse olfactory bulb tissue captured using the Slide-seqV2 platform (Wang et al., 2022b). Each slice contains 131,828 × 131,828 spatial data points, with detailed gene expression profiles. For training, slices 1, 3, 5, 6, and 8 were selected, while slices 2, 4, and 7 were reserved for testing. This setup allows us to evaluate both intra-slice and inter-slice reconstruction capabilities in a highly spatially resolved dataset.

**Human Dorsolateral Prefrontal Cortex (DLPFC) Dataset**. The DLPFC dataset consists of four slices of human dorsolateral prefrontal cortex tissue obtained using the 10x Visium platform (Maynard et al., 2021). Each slice contains 3,460 spatial data points, where each point represents a 3,000-dimensional gene expression profile. To assess intra-slice reconstruction performance, we randomly removed 50% of the spatial data points from a single slice. The remaining 50% were used for training, and the complete set of points was used for testing.

**Drosophila Embryo (DE) Dataset**. This dataset includes 12 slices of a Drosophila embryo at the 16–18 hour developmental stage, captured with spatial transcriptomic techniques (Wang et al., 2022c). This dataset serves as an additional validation to test the model's robustness in capturing spatial continuity across developmental stages.

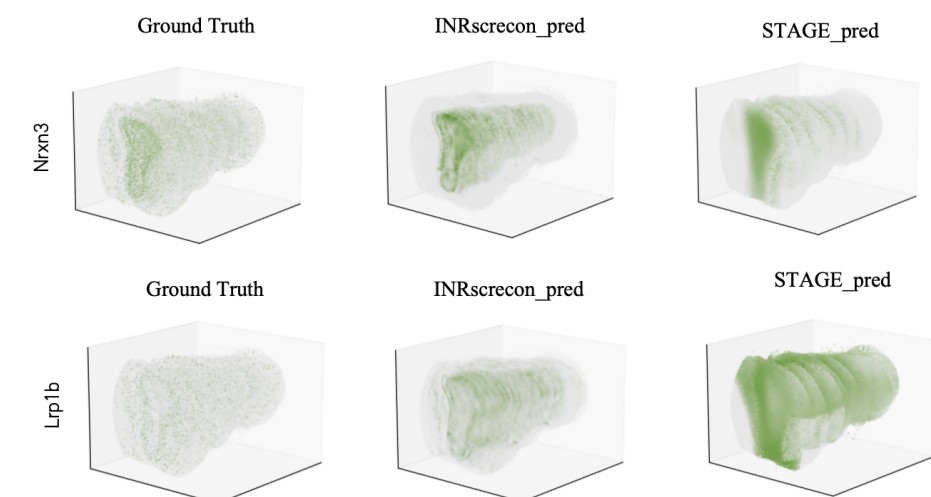

**Figure 4: Continuous Spatial Distribution of Genes** This figure demonstrates the recovery of spatial gene distribution on missing sections, comparing the results between the STAGE and INRscrecon models. The ground truth represents the 3D distribution of a gene, and in our predictions, we aim to observe this pattern.

These datasets were chosen to ensure a broad evaluation of INRscrecon under diverse biological and technical conditions. They enable us to systematically analyze its ability to handle both **Intra-slice Completion** (restoring missing data within a single slice) and **Inter-slice Prediction** (reconstructing data across multiple slices). By leveraging these datasets, we aim to rigorously validate the model's ability to reconstruct spatially continuous 3D gene expression profiles.

In terms of 3D reconstruction technologies, we employed two advanced methods as baselines: STitch3D and STAGE. STitch3D utilizes GCNs to precisely restore each spatial position, widely regarded as an effective technique for addressing spatial localization in spatial transcriptomics data. STAGE, with its advanced encoder-decoder architecture, effectively supplements missing parts in the data, demonstrating unique advantages in handling dynamic spatial data. By comparing these two technologies, we can not only evaluate the clarity and accuracy of our reconstruction results but also enhance our understanding of complex biological tissue structures and functions. All model training and evaluations were conducted on a personal computer equipped with an Intel i7-9700 CPU at 3.00 GHz, 32 GB of RAM, and an NVIDIA GeForce RTX 4090 GPU.

### 4.2 QUANTITATIVE AND QUALITATIVE 3D RECONSTRUCTION RESULTS

For quantitative evaluation, we employed three datasets with original annotations as ground truth references. To assess the accuracy of our models, we applied the Adjusted Rand Index (ARI) (Yuan et al., 2024) and the Structural Similarity Index Measure (SSIM) (Li et al., 2022). These metrics are widely recognized for their efficacy in evaluating spatial transcriptomics algorithms. As detailed in Table 1, our INRscrecon shows good performance than existing techniques across all evaluated datasets, thereby demonstrating its superior ability to reconstruct 3D gene profiles.

To demonstrate the reconstruction capabilities of INRscrecon, we conducted detailed analyses across three distinct datasets. Firstly, for the OB dataset, we restored gene expression levels from missing sections, comparing the performance of INRscrecon with the STAGE method. First, as shown in Figures 3 and 4, our method not only guarantees the continuity of 3D gene restoration but also ensures consistent recovery of gene expression from individual genes. In the images, GT represents the distribution of gene expression levels. By analyzing data extracted with similar expression levels, we observe that the expression levels display similar distribution ratios, both in two-dimensional and three-dimensional representations, thereby demonstrating the robust performance of our method across different dimensions.

Secondly, in the analysis of the DLPFC dataset, we focused on reconstructing layer data from removed spots, also compared with STAGE, as shown in Figure 5. We randomly remove 50% points, and use the remaining 50% as training sets, INRscrecon demonstrated superior performance in re-

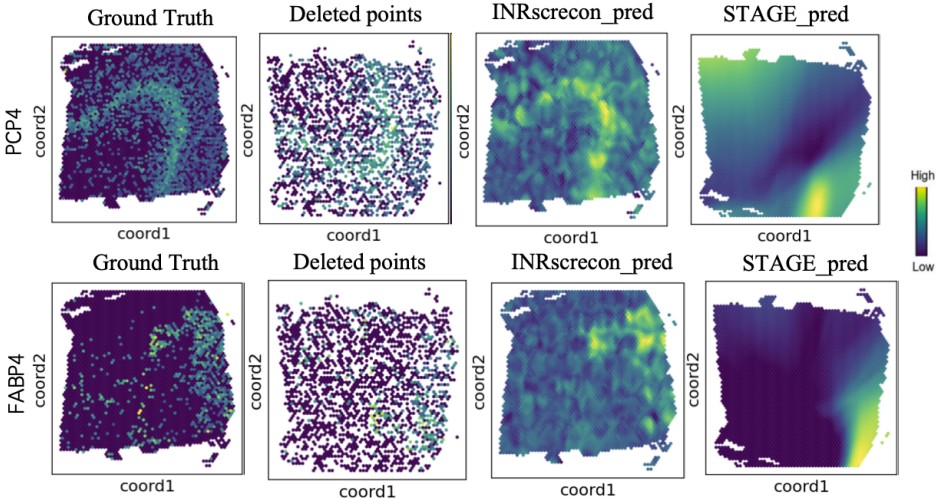

**Figure 5: Results After Random Spot Deletion** Panels display the spatial distribution of gene expression for two genes across four conditions: ground truth, deleted points, INRscrecon predictions, and STAGE predictions. The color gradient from blue to yellow indicates low to high gene expression levels, respectively. This comparison highlights the differences in the ability of each method to reconstruct complex spatial gene expressions accurately.

covering the deleted points. This underscores INRscrecon's enhanced ability to accurately reconstruct spatial data under conditions of partial data loss. We conducted biological downstream analyses, including cell type annotation, with promising results detailed in appendix.

Lastly, in our experiments with the DE dataset, we evaluated the performance of the conducted biological downstream analyses in 3D, including cell type annotation in 3D, using the INRscrecon model by examining its ability to reconstruct 3D cell types. Comparisons with the original DE data annotations (Figure 6A) showed that the foregut and midgut domains reconstructed by our model closely resembled the original annotations (Figure 6B), where the blue and green dots indicate different cell types in the spatial regions. This demonstrates INRscrecon's exceptional ability to capture intricate details within these regions. The Pearson correlation coefficient between the reconstructed and original annotations was significant at 0.71, quantitatively confirming the visual similarity observed.

Overall, INRscrecon not only displays good capabilities in reconstructing continuous 3D profiles but also provides clearer views for observing and analyzing 3D cellular structures. This is particularly crucial when dealing with highly heterogeneous and spatially uneven biological data, demonstrating that INRscrecon can effectively be used for complex spatial transcriptomics data analysis.

**Table 1:** Comparative Performance of Different Methods on Various Datasets

| Dataset | Method | ARI | SSIM |
|---|---|---|---|
| OB(Wang et al., 2022b) | STitch3D | 0.50 | 0.66 |
| | STAGE | 0.30 | 0.51 |
| | INRscrecon | 0.72 | 0.73 |
| DLPFC(Maynard et al., 2021) | STitch3D | 0.46 | 0.55 |
| | STAGE | 0.33 | 0.45 |
| | INRscrecon | 0.65 | 0.70 |
| DE(Wang et al., 2022c) | STitch3D | 0.49 | 0.66 |
| | STAGE | 0.44 | 0.36 |
| | INRscrecon | 0.60 | 0.68 |

## 4.3 ABLATION STUDY FOR MODEL PERFORMANCE

To investigate the reasons behind our model's superior performance, we utilized OB dataset to dissect the functionality of each component of our model's loss function. Our analyses demonstrate that incorporating representation consistenc(RC) loss and KL divergence loss significantly enhances

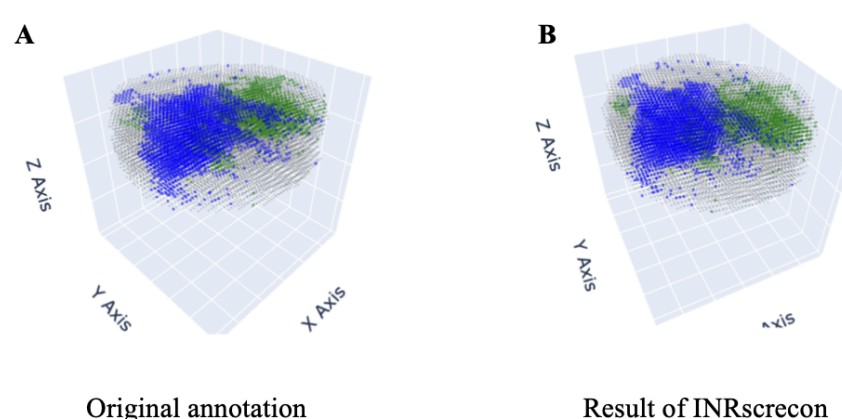

Original annotation            Result of INRscrecon

**Figure 6: 3D Reconstruction of Foregut and Midgut Domains** Panel A shows the original annotation of cell types in the DE dataset displayed in a 3D plot, illustrating the spatial distribution of foregut and midgut domains. Panel B presents the result of the INRscrecon model's reconstruction of the same dataset. Blue and green dots indicate different cell types within the spatial regions, highlighting the model's ability to closely mimic the original annotations and capture intricate spatial details. The comparison underscores INRscrecon's exceptional performance in accurately reconstructing complex 3D cell types.

model performance. Through both qualitative and quantitative assessments, we observed continuous improvements in our model's outputs as adjustments were made(Figure 7, Table 2). Notably, the INRscrecon model consistently outperformed others, delivering the most accurate results in terms of both spatial and gene expression fidelity.

## 5 DISSCUSION

Our study demonstrates the robust capabilities of the INRscrecon model within spatial transcriptomics, revealing its significant potential for biological microscale applications. The superior performance of INRscrecon across various datasets confirms its effectiveness in accurately reconstructing complex 3D gene profiles. This precision is critical not only for advancing our understanding of cellular architectures but also for facilitating detailed exploration of spatial and genetic interactions within tissues.

The transition from two-dimensional to three-dimensional spatial transcriptomics has been greatly enhanced by computational techniques like Neural Radiance Fields (NeRF) (Mildenhall et al., 2021) and 3D Gaussian Splatting (GS)(Kerbl et al., 2023), which have proven essential in reconstructing intricate biological structures at the microscale. These advancements contribute profoundly to our capacity to visualize and analyze the spatial organization of cells and their gene expression in unprecedented detail.

Looking forward, the integration of INRs with these advanced computational models opens new avenues for precision medicine. By improving the granularity and accuracy of spatial transcriptomics, we can better align technological progress with clinical needs, particularly in the development and optimization of personalized treatment strategies. The potential to tailor therapeutic approaches based on detailed 3D mappings of tissue architecture promises a new era of treatment customization, enhancing patient outcomes in complex diseases.

As research delves deeper into the microscopic aspects of biological phenomena, the ongoing refinement of INRs and related technologies will play a crucial role in the future of biomedical research, offering new insights and tools for scientists and clinicians alike. The application of these sophisticated computational tools in routine clinical practice could revolutionize how we understand and treat various diseases, marking a significant step forward in the integration of bioinformatics and healthcare.

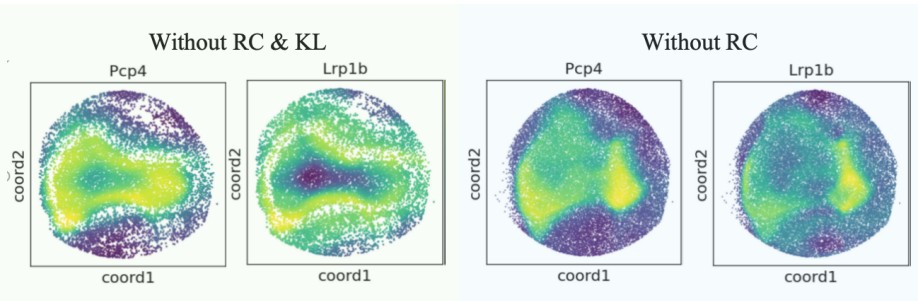

**Figure 7:** Performance Improvements with Losses

**Table 2:** Ablation Study Results

| Dataset | Method | ARI | SSIM |
|---|---|---|---|
| OB | w/o RC loss and KL loss | 0.22 | 0.27 |
| Wang et al. (2022b) | w/o RC loss | 0.33 | 0.45 |
| | w/ losses | 0.65 | 0.70 |

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
