# APPENDIX

## 1 BACKGROUND MATERIALS ABOUT SPATIAL TRANSCRIPTOMICS

The rapid development of spatial transcriptomics has significantly advanced the study of tissue structures and the spatial distribution of gene expression. By integrating high-throughput sequencing, barcode labeling, and high-resolution imaging technologies, researchers can combine spatial information across serial sections to achieve 3D reconstruction. This methodology not only supports comprehensive analysis of tissue expression patterns but also shows great potential in cancer research, neuroscience, and multi-organ development studies.The strategy begins with sample preparation and sectioning. Embedded tissue samples are processed into ultra-thin, continuous sections using high-precision microtomes, preserving the spatial integrity and consistency between sections (Li & Peng, 2022). Each section retains the spatial structure and local gene expression characteristics of the tissue, providing a solid foundation for subsequent data acquisition and analysis. In the data acquisition phase, high-throughput sequencing technology combined with spatial barcodes (spatial barcode/ID) links gene expression data to their spatial coordinates, generating transcriptomic data with spatial characteristics (Waylen et al., 2020). Additionally, high-resolution imaging techniques, including in situ sequencing and in situ hybridization, precisely localize gene expression patterns at the molecular level within each section. The integration of these multimodal data greatly enhances the resolution and accuracy of 3D reconstruction. Through data integration and 3D modeling, researchers can generate a 3D gene expression map of the tissue. Rao (Rao et al., 2021) demonstrated how the dynamic distribution of gene expression in 3D space reveals the functional partitioning of different cell populations and their interactions. 3D reconstruction not only restores the original morphology of the tissue but also uncovers the mechanisms underlying diseases and regulatory processes within the cellular microenvironment. This 3D reconstruction strategy based on spatial transcriptomics helps researchers achieve a more comprehensive understanding of the gene expression patterns in complex tissues. It also provides robust tools for disease studies and tissue engineering. The continuous refinement of these technologies is expected to open new frontiers for biological research.

## 2 CNN ARCHITECTURE

The CNN architecture consists of two convolutional layers, followed by a max pooling layer, and two fully connected layers with ReLU activation functions applied between them.

- Input Dimension: 5,000 (gene expression features for each sample).

- Output Dimension: 128 (learned feature embedding for downstream tasks).

- Conv1: A 1D convolutional layer with an input channel size of 1 and output channel size of 256 (hidden dim). The kernel size is 3, with padding of 1 to maintain sequence length.

- Conv2: Another 1D convolutional layer with the same configuration as Conv1. Pooling

- Max-pooling layers with a kernel size of 2 are applied after each convolutional layer to reduce the sequence length, effectively halving it at each step.

- FC1: A fully connected layer mapping the flattened features to a hidden dimension of 512.

- FC2: The final fully connected layer projects the 512-dimensional features to the output dimension of 128.

- ReLU activation is applied after each convolutional and fully connected layer to introduce non-linearity.

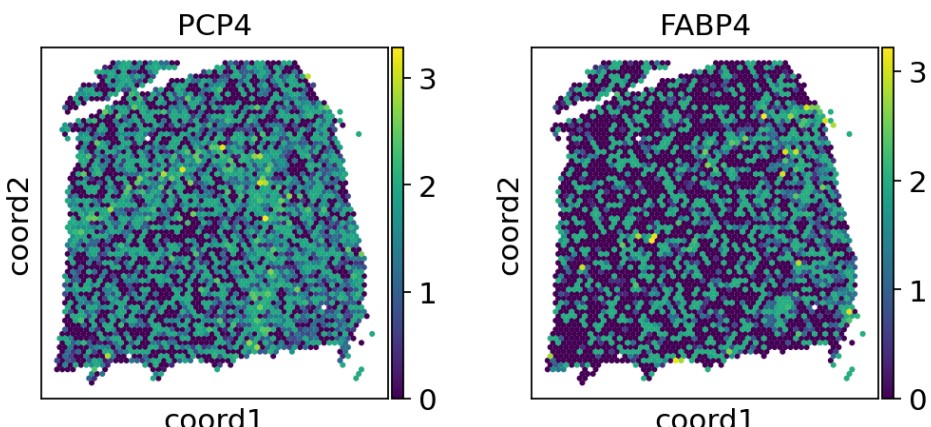

Figure 1: **Reconstruction results of cubic interpolation.**

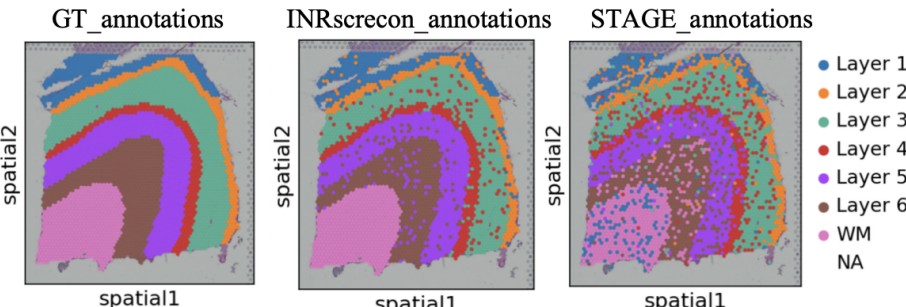

Figure 2: **Comparison of Spatial Annotations for Different Methods** This figure illustrates the annotation results for a spatial dataset, showing the distribution of different tissue layers across three panels: GT_annotations (Ground Truth), INRscrecon_annotations, and STAGE_annotations. Each color represents a distinct layer or category: Layer 1 (blue), Layer 2 (orange), Layer 3 (green), Layer 4 (red), Layer 5 (purple), Layer 6 (light blue), White Matter (WM, pink), and Not Applicable (NA, gray). The GT panel provides the original annotations as a reference, while the INRscrecon and STAGE panels display the respective methods' ability to replicate or improve upon these annotations. The figures highlight the variations and accuracy of each method in capturing the spatial distribution and layer segmentation within the dataset.