# OpenReview forum: "INRscrecon: Enhancing 3D Spatial Transcriptomics Reconstruction through Implicit Neural Representations"
_ICLR.cc/2025/Conference — Submitted to ICLR 2025_

### Official Review · Reviewer_RbdE · 2024-11-03

**Soundness:** 1
**Presentation:** 1
**Contribution:** 2
**Rating:** 1
**Confidence:** 4

**Summary:**

This paper introduces an INR-based approach with latent feature alignment for single-cell spatial transcriptomics rigid alignment. It employs INRs in the context of reconstruction and CNNs for latent space alignment, utilizing a combination of three different loss functions. In the paper, the authors compare the presented method against two state-of-the-art baselines, STitch3D and STAGE, outperforming in the reconstruction task in terms of SSIM/ARI on three datasets, namely OB, DLPFC, and DE.

**Strengths:**

Originality: To the best of my knowledge, this paper is among the first to explore INRs in the context of (spatial) transcriptomics. Moreover, it uses an interesting combination of INRs, which operate on the coordinate level and continuously model the information with fully connected MLPs, and a feature alignment model with CNNs that operate in the image domain. This combination is relatively unique, especially in this context, and presents an interesting idea.

Significance: Given the new and interesting application of INRs in the transcriptomic domain, I believe the presented method would be relevant to the biological modeling community and may also spark interest in other INR-based modeling approaches, given that the learning-based feature alignment may constitute a generalized concept applicable for (rigid) registration.

Quality: Considering the flaws in presentation and lack of details/concerns in evaluation, it is difficult to mention particular strengths of the paper in terms of its quality.

Clarity: The authors describe their problem statement very clearly (in the introduction).

**Weaknesses:**

Concerns regarding fair and optimal evaluation:

The manuscript states that all training was "limited to a duration of 30 minutes", which if I understand it correctly, does not consider - at all - if the models for the baseline (nor the presented method) have converged. Given this statement, it is highly unlikely the hyperparameters were optimized for the baselines, and I would doubt if the model performance of the presented method has been optimized as well in terms of hyperparameters. Given that the authors fail to provide any further details, the evaluation is very concerning and lacks any chance of reproducibility.

Lack of clarity in writing:

Abstract: Given the length and content, I believe the Abstract does not reflect the actual paper content. It does not precisely state the problem the paper aims to address and does not mention the experimental setup and results. It is, for example, not clear that the authors use CNN-based feature alignment (at all), nor is it clear for the task of rigid (!) alignment.

Introduction & Related Work:
- While I believe none of the content is wrong, substantial parts of related work and introduction are not ultimately relevant to the presented method.
- While INRs have become prominent in MRI and CT reconstruction applications, this paper presents a very different application. The authors may acknowledge the work in other domains, but they should focus on more similar work and set their work into this context. For instance, INRs have been applied to cells [1] and non-deformable registration [2] - works that are ultimately (more) relevant to this paper but absent from the related works section. Also, I believe it would be very important to differentiate between cohort-based and single-instance approaches (two common INR concepts) and technical advances in INRs (e.g., SIREN, Hashgrid Encoding, Meta-Learning, etc.).

Discussion: The discussion does not **discuss** any relevant parts of the experiments. It does not provide any intuition on why the presented method provides better performance or why baselines fail or perform considerably worse. Moreover, the authors do not discuss limitations, etc., but just introduce new papers that are not relevant to the discussion (i.e., NeRFs). This section really needs re-writing.

Experimental design, hyperparameters, and training setup:

- The code is not available. Thus, it is ultimately important to have a detailed overview and insight into the architectural parameters for the paper, which is absent (except 3.2.2. regarding the number of layers in the MLP).
- The authors do not state any hyperparameters of the baselines and selected method. It is unclear if the evaluation is fair and if the selected method was well-tuned.
The authors do not mention how the training scenarios differ for the baselines given that (mostly) only INR takes a single-image approach - were they trained (at all?), and were they trained in a fair manner?
- The authors use positional encoding as in [3] but fail to appropriately cite it. Moreover, the reconstruction results look very blurry - perhaps this points to an issue in selecting appropriate Fourier Features [3], which have to be tuned and tailored to the application. Perhaps Hash-grid encoding would have been a better encoding? [4]
- Experiments do not feature standard variations for INRscrecon and baselines (even though the header indicates that the authors wanted to do so), offering limited insights into the robustness of the method.
- The authors show the results of ablation experiments but do not state how the final model weighted the different loss functions.

Conclusion:
In its current form, I believe the paper has major issues - especially in related works, reproducibility, and baseline comparisons, that do not comply with the standards posed by ICLR. The authors should incorporate the feedback into a revised version of the manuscript.

References:
[1] Wiesner D, Suk J, Dummer S, Svoboda D, Wolterink JM. Implicit neural representations for generative modeling of living cell shapes. InInternational Conference on Medical Image Computing and Computer-Assisted Intervention 2022 Sep 16 (pp. 58-67). Cham: Springer Nature Switzerland.

[2] Wolterink JM, Zwienenberg JC, Brune C. Implicit neural representations for deformable image registration. InInternational Conference on Medical Imaging with Deep Learning 2022 Dec 4 (pp. 1349-1359). PMLR.

[3] Tancik M, Srinivasan P, Mildenhall B, Fridovich-Keil S, Raghavan N, Singhal U, Ramamoorthi R, Barron J, Ng R. Fourier features let networks learn high-frequency functions in low dimensional domains. Advances in neural information processing systems. 2020;33:7537-47.

[4] Müller T, Evans A, Schied C, Keller A. Instant neural graphics primitives with a multiresolution hash encoding. ACM transactions on graphics (TOG). 2022 Jul 22;41(4):1-5.

**Questions:**

Questions - Baselines and Experiments
- How were the baselines trained? Please specify train-/val-/test splits.
- Were baselines fine-tuned? Was a hyperparameter search conducted?
- How was the feature CNN trained? What is it's architecture (layers, dropout, kernel, ..)
- Was the CNN trained on single images/stacks or multiple ones? Was it pre-trained with other datasets?
- How did you ensure a fair evaluation of baselines compared to the selected method?

Suggestions:
Please consider the weaknesses highlighted in the relevant section. I believe the authors should address the important concerns regarding the (fair) evaluation and work on the method presentation.

---

### Official Review · Reviewer_kFhC · 2024-11-03

**Soundness:** 1
**Presentation:** 2
**Contribution:** 2
**Rating:** 5
**Confidence:** 2

**Summary:**

This paper proposes INRscrecon, a new method based on implicit neural representation (INR) to improve 3D spatial transcriptomics reconstruction. The proposed INRscrecon method includes two stages: 1) Alignment, where two spatial coordinates from single-cell data are registered using a rigid transformation, and 2) Reconstruction, where an INR model is trained to represent gene expression profiles. By leveraging the continuous prior provided by the INR model, this approach achieves improved 3D spatial transcriptomics reconstruction.

**Strengths:**

**Motivation:** This work attempts to address the important problem of 3D spatial transcriptomics reconstruction using an INR model.

**Clarity and organization:** This paper is well-written and easy to follow.

**Weaknesses:**

**Technical soundness**: Many studies have demonstrated that the continuous prior inherent in the INR network (also known as spectral bias [4]) serves as a powerful and general regularization for various inverse problems, including novel view synthesis [1][2], medical reconstruction [3], and more. However, most INR-based methods follow a standard framework: 1) using the INR network to represent the signals to be reconstructed; 2) using a differentiable forward model to simulate the physical acquisition process; 3) optimizing the INR network by minimizing the prediction errors on measurement data. The success of these methods largely relies on these aspects (i.e., the INR prior and the physical model). However, the proposed INRscrecon model directly uses an MLP network to fit the gene expression data without effectively modeling the physical acquisition process. In my opinion, this approach may not perform significantly better than traditional interpolation methods, such as cubic interpolation.

**Experimental results**: From Figure 5, the reconstructions by the INRscrecon model appear to deviate significantly from the GT. Please clarify this observation.   Additionally, since the proposed method resembles traditional interpolation methods in essence, please provide comparison results using interpolation methods.

**Area of expertise**: For the ICLR audience, 3D spatial transcriptomics reconstruction may be an unfamiliar field. Providing additional background information would improve readability. Given the nature of this work, it may be more suited to bioinformatics venues, such as ISMB and ECCB.

> [1] Mildenhall, Ben, et al. "Nerf: Representing scenes as neural radiance fields for view synthesis." Communications of the ACM 65.1 (2021): 99-106.

> [2] Pumarola A, Corona E, Pons-Moll G, et al. D-nerf: Neural radiance fields for dynamic scenes[C]//Proceedings of the IEEE/CVF Conference on Computer Vision and Pattern Recognition. 2021: 10318-10327.

> [3] Molaei, Amirali, et al. "Implicit neural representation in medical imaging: A comparative survey." Proceedings of the IEEE/CVF International Conference on Computer Vision. 2023.

> [4] Rahaman, Nasim, et al. "On the spectral bias of neural networks." International conference on machine learning. PMLR, 2019.

**Questions:**

See the section of Strengths and Weaknesses, please.

---

### Official Review · Reviewer_Q6SK · 2024-11-04

**Soundness:** 2
**Presentation:** 2
**Contribution:** 3
**Rating:** 5
**Confidence:** 2

**Summary:**

This paper introduces INRscrecon, which utilizes INRs to precisely predict and correct missing and distorted data sections, thereby enhancing the clarity and accuracy of tissue 3D reconstructions. This approach effectively addresses misalignments and data inconsistencies inherent in traditional experimental setups, markedly refining the fidelity of 3D spatial transcriptomic reconstructions.

**Strengths:**

1. This paper focuses on a realistic scientific problem.
2. The use of INR is a good choice for imputation.
3. The literature review is good.

**Weaknesses:**

1. The method and data description is a bit confusing. How is location-related information encoded in gene expression profile? Why can one use CNN to process genes? Does neighborhood numbers in $G$ represent spatially-close information?
2. It is not clearly illustrated why the data is distorted. The experiment part does not show how well this approach fixes the distortion compared to another approach.
3. The visualizations of gene expressions are confusing. What is the purpose of this visualization?
4. There are just a few slices in each dataset. It is worried that the results in Table 1 are insufficient to show the approach's effectiveness. Also, there are no variances/standard deviations/p-values of the evaluation metrics.
5. How to understand Figure 6? What do the blue and green dots mean?
6. Although the benefit of INR is clearly stated, the paper did not give the unique actual application of this approach because of the benefit of INR.

**Questions:**

1. There is a lack of essential details to understand the paper. How big are the datasets? How are the training and testing sets split? Is one network for one slice? Or is the network responsible for representing information from all slices?
2. How many points $C$ and gene expression profiles $G$ are used?
3. How to choose the $\lambda1$, $\lambda2$ and $\lambda3$ in Equation 7?

---

### Official Review · Reviewer_EtyP · 2024-11-08

**Soundness:** 2
**Presentation:** 2
**Contribution:** 2
**Rating:** 3
**Confidence:** 4

**Summary:**

This paper presents a method using INRs to continuously encode signals from single-cell spatial transcriptomics (scST). Specifically, they focus on the problem of 3D tissue reconstruction. Achieving accurate profiles of scST in 3D tissue is rendered challenging due to high likelihood of data contamination. Consequently, existing methods are forced to rely on low-resolution discrete representations to model the continuous representations of the data. The authors present a 3D reconstruction algorithm using INRs to impute missing gene expression data. They demonstrate results on 3 datasets and compare with 2 benchmark algorithms.

**Strengths:**

This work highlights an interesting and relevant problem in biology: reconstruction of 3D single-cell spatial transcriptomics. The authors clearly outline issues with the current practice of obtaining said data, and limitations in using a discrete representation. The authors propose an INR to improve the signal in spatial cell recognition and gene expression recovery. While they use standard loss functions and a standard framework for reconstruction, they demonstrate significantly improved performance on three tasks compared with 2 baseline methods.

**Weaknesses:**

While the paper addresses an important problem in 3D scST, this paper lacks methodological novelty and rigorous evaluation.

### Method
The method is comprised of many standard blocks in 3D geometry. The alignment problem is simply a modification of the ICP algorithm. The INR model uses standard building blocks such as positional encodings and standard reconstruction losses. However, the authors do propose a representation consistency loss, which aims to map the input features to a latent space. The block maps the real and predicted features to the same space to ensure consistency. This loss appears to have a significant effect on performance as demonstrated by the ablation. However, this is not a novel proposed loss.

Lacking methodological novelty is fine provided the paper were accompanied by extensive experimentation on real, challenging data. However, the experiments are also limited, and are done on synthetic tasks.

### Experiments
There are many questions and points of uncertainty in the experimental setup. As described, the authors use 3 datasets and quantify the ability of their method to continuously reconstruct 3D scST profiling. From these datasets, the authors randomly removed sections or points as data to reconstruct. However, there are many questions about this procedure and its reproducibility. First, why is removing sections or points a realistic experimental evaluation scenario? How was the random selection done? The authors should have used and described a reproducible criteria for this.

From the baselines, it is unclear why these baselines are proper models to compare against. Why is a GCN model used for signal reconstruction? Why also use a simple VAE model rather than a more sophisticated one? Most importantly, why was training session limited to a duration of 30 minutes? This likely handicapped the baselines. Also, how was the data split? Was there a train and validation set? How were these used for the baselines?

Finally, I have a hard time interpreting the results, particularly in the figures. The figure coordinate scales are not consistent nor described. In Fig. 7, why are there such large differences in the plotted color values? I also find the other figures difficult to interpret and are missing text describing the main takeaways.

### Clarity of presentation
- There are typos throughout the paper
- Many figures are not descriptive and are missing proper legends. For example, Figure 1 has too much information with minimal accompanying text. The colormaps across figures are not consistent. For example the scale on Figure 3 is twice as large for the real data than the predicted.

**Questions:**

Many of my questions may be found in the section above.

---

### Meta-Review · Area_Chair_BpZ2 · 2024-12-22

**Metareview:**

The paper present INRscrecon, a framework leveraging Implicit Neural Representations (INRs) to address the challenges of 3D tissue reconstruction in single-cell spatial transcriptomics (scST). By encoding continuous signals, INRscrecon overcomes limitations of low-resolution discrete methods, expecting to get more accurate prediction and correction of missing or distorted gene expression data. This method includes two stages: rigid alignment of spatial coordinates using latent feature alignment and INR-based reconstruction of gene expression profiles with a combination of loss functions. Tested on three datasets, INRscrecon compares with two baselines methods (STitch3D and STAGE) in reconstruction fidelity.

Strength: This paper addresses a significant and realistic problem of 3D single-cell spatial transcriptomics reconstruction, highlighting limitations in current practices that rely on discrete representations. By leveraging Implicit Neural Representations (INRs) for spatial cell recognition and gene expression recovery, the proposed method demonstrates substantially improved performance across three tasks compared to two baseline methods. The interesting application of INRs in the transcriptomic domain presents a relevant contribution to biological modeling, offering a generalized approach to feature alignment and rigid registration that may inspire further exploration in INR-based methods.

Weakness: Most reviewers think the paper lacks the methodological novelty and rigorous evaluation and baseline comparison. The experiments were limited to synthetic tasks and an unclear experimental setup. Some presentations of results are difficult to interpret, and some descriptions of the method and data may be confusing. The discussion section does not provide insights into the method’s performance or its comparison with baselines, while also overlooking limitations. Additionally, issues with related work coverage, reproducibility, and baseline comparisons may undermine the paper's overall rigor.

Overall, considering the paper’s contribution and the remaining concerns, I suggest rejection.

**Additional Comments On Reviewer Discussion:**

During the rebuttal period, most reviewers have responded to the author's rebuttal and follow up. While some concerns are well addressed in the rebuttal, reviewers also pointed out that some important concerns are neglected by the author's rebuttal, including baseline comparison setting, related work, experimental discussion, and the discussion.

---

### Decision · Program_Chairs · 2025-01-22

Reject